# Enterotoxigenic and Antimicrobic Susceptibility Profile of *Staphylococcus aureus* Isolates from Fresh Cheese in Croatia

**DOI:** 10.3390/microorganisms11122993

**Published:** 2023-12-15

**Authors:** Ivana Ljevaković-Musladin, Lidija Kozačinski, Marija Krilanović, Marina Vodnica Martucci, Mato Lakić, Luca Grispoldi, Beniamino T. Cenci-Goga

**Affiliations:** 1Environmental Health Department, Public Health Institute of Dubrovnik-Neretva County, Dr. A. Šercera 4A, HR-20000 Dubrovnik, Croatia; mato.lakic@zzjzdnz.hr; 2Department of Hygiene, Technology and Food Safety, Faculty of Veterinary Medicine University of Zagreb, Heinzelova 55, HR-10000 Zagreb, Croatia; klidija@vef.unizg.hr; 3Microbiology Department, Public Health Institute of Dubrovnik-Neretva County, Dr. A. Šercera 2C, HR-20000 Dubrovnik, Croatia; marija.krilanovic@zzjzdnz.hr (M.K.); marina.vodnica-martucci@zzjzdnz.hr (M.V.M.); 4Laboratorio di Ispezione Degli Alimenti di Origine Animale, Department of Veterinary Medicine, University of Perugia, 06126 Perugia, Italy; grisluca@outlook.it (L.G.); beniamino.cencigoga@unipg.it (B.T.C.-G.)

**Keywords:** *Staphylococcus aureus*, cheese, enterotoxigenicity, staphylococcal enterotoxins, antimicrobic susceptibility

## Abstract

Certain *Staphylococcus aureus* strains harbour staphylococcal enterotoxin genes and hence can produce enterotoxin during their growth in food. Therefore, food can be a source of staphylococcal food poisoning, one of the most common food-borne diseases worldwide. Epidemiological data show that *S. aureus* is often present in raw milk cheeses, and consequently, cheeses are often the source of staphylococcal food poisoning outbreaks. The aim of this study was to determine the phenotypic characteristics of *S. aureus* isolates from fresh cheese, including antibiotic susceptibility; the presence of classical sea-see enterotoxin genes through molecular methods; and the isolate’s ability to produce SEA-SEE enterotoxins in vitro through reversed passive latex agglutination. A total of 180 coagulase-positive staphylococci were isolated from 18 out of 30 cheese samples, and 175 were confirmed as *S. aureus* through latex agglutination and API STAPH tests. All isolates possessed phenotypic characteristics typical for *S. aureus*, with certain variations in the egg yolk reaction (18.3% of the isolates showed a weak reaction and 28% no reaction at all) and haemolysis pattern (36.6% of the isolates produced double-haemolysis and 4.6% were non-haemolytic). Antibiotic resistance was observed in 1.1% of the isolates and to mupirocin only. Real-time PCR detected the *sec* gene in 34 (19.4%) isolates, but most isolates (80.6%) were not enterotoxigenic. For all 34 (19.4%) strains that carried the *sec* gene, the RPLA method detected the production of the SEC enterotoxin in vitro. For those enterotoxigenic strains, the possibility of enterotoxin production in fresh cheese could not be ruled out.

## 1. Introduction

Staphylococcal food poisoning (SFP) is one of the most common food-borne diseases worldwide. The causative agents are enterotoxins produced by enterotoxigenic strains of *S. aureus* during its growth in favourable conditions in food. The most commonly reported foods involved in SFP outbreaks in the EU are mixed foods (29.7%), meat and meat products (20.8%), cheese and dairy products (14.4%), bakery products (8.4%), and fish and fish products (6.5%) [1,2]. According to De Buyser et al. [3], *S. aureus* is the main pathogen associated with raw milk cheeses. Delmas et al. [4] claim that *S. aureus* is the foremost causative agent of food poisoning from milk and dairy products. A meta-analysis conducted by Gajewska et al. [5] on studies during the years 2011–2021 reported that *S. aureus* was isolated from 29.07% of the milk samples and 42.81% of the cheese samples. The authors concluded that, despite food safety criteria for *S. aureus* that exist worldwide, the prevalence of this pathogen in raw milk and raw milk cheeses is very high. Mastitic cows are considered the most important source of milk contamination by enterotoxin-producing *S. aureus* strains [6,7,8]. Staphylococcal enterotoxins (SEs) are a superfamily of 29 exotoxins and are one of the major *S. aureus* virulence factors. The SE family is divided into two groups: classical and newer enterotoxins. At present, 29 kinds of enterotoxins have been found, which can be divided into classic enterotoxins (including SEA-SEE), non-classical enterotoxins (such as SEG, SEH, etc.), and enterotoxin-like (such as SElW, SElX, etc.) [9].

These multifunctional proteins share the same structural and functional properties, as well as the same superantigen activity [10]. Classical SEs were first described back in 1960 due to their emetic activity and implications in food poisoning outbreaks [11]. The role of newer SEs in staphylococcal food poisoning has yet to be proven. Epidemiological data show that SEA is the most frequently found SE in SFP outbreaks, followed by SED, SEB, and SEC, while SEE is the rarest [11,12,13]. Genotyping studies of *S. aureus* isolates from food implicated in staphylococcal food poisoning also confirm the prevalence of *sea*, *sed*, and *seb* genes [14,15,16,17,18,19].

Many studies have shown that *S. aureus* isolates from cheese carry one or more SE genes [20,21,22,23,24]. Published data show remarkable variability in the frequency and diversity of the prevalence of enterotoxin genes worldwide. Earlier studies showed the prevalence of *sec*, *sea*, and *sed* genes in *S. aureus* isolates [14,15,16,17,18,20,21,22,23,25,26,27,28,29,30,31,32,33,34,35,36,37,38,39,40,41]. Recent studies showed a high frequency of newer enterotoxins [15,20,21,22,23,42,43,44,45]. The most recent systematic review and meta-analysis on the prevalence, antibiotic resistance, and enterotoxin genes of *S. aureus* isolated from milk and dairy products worldwide, conducted by Zhang et al. [46], reported that the pooled rate of classical staphylococcal enterotoxins was 39.31% (95% CI: 25.99–53.44%), and the highest rates were found for *sec* (16.27%; 95% CI: 10.32%–23.15%) and *sea* (9.11%; 95% CI: 3.81%–16.39%) genes.

*S. aureus* represents a serious hazard for the end consumer, as it is able to produce enterotoxins, which are stable at high temperatures (e.g., crude enterotoxin A remains active at 100 °C for 2 h in broth and at 121 °C for 28 min in mushrooms) and are also resistant to many environmental conditions (low pH, freezing, drying), which *S. aureus* strains do not survive. They are also resistant to human proteolytic enzymes and retain their activity in the digestive tract after ingestion [47]. The amount of enterotoxin required to cause illness in susceptible subjects can be as little as 20–100 ng [48]. A fraction of the strains of *S. aureus* are also able to persist in factory environments by forming biofilm [49].

A previous study conducted by Ljevaković-Musladin et al. [50] showed that home-made cheeses from the markets of the city of Dubrovnik had a high frequency and high level of contamination with *S. aureus*, indicating the need for further investigation of the enterotoxigenicity of isolates and assessment of the real risk of staphylococcal poisoning.

Since there are no data on the nature of *S. aureus* isolated from cheese produced in Croatia, the aim of this study is to determine (i) the phenotypic characteristics of isolates, including their antibiotic susceptibility; (ii) the presence of classical sea-see enterotoxin genes through molecular methods; and (iii) the ability of isolates to produce SEA-SEE enterotoxins in vitro, as detected through reversed passive latex agglutination (RPLA).

## 2. Materials and Methods

**Cheese samples.** A total of 30 fresh cheese samples produced by small-scale plants (26 bovine, 2 caprine, and 2 bovine/caprine) were collected from the Dubrovnik city markets during 2018 and 2019. For each sampling day, cheeses were randomly chosen from the plant database: all cheese numbers were printed, cut out, and then randomly drawn from a hat.

**Microbiological analyses of cheeses.** All cheese samples were analysed according to HRN EN ISO 6888-1:2004 [51] using Baird-Parker agar for the enumeration of coagulase-positive staphylococci. Ten grams of cheese was homogenised with 90 mL of Buffered Peptone Water (Oxoid, Basingstoke, UK) in a Smasher^®^ blender (bioMerieux, Marcy-l’Etoile, France). Then, 0.1 mL of ten-fold dilutions was inoculated in duplicate on Baird-Parker agar (Oxoid, Basingstoke, UK) and incubated at 37 ± 1 °C for 24–48 h. After incubation, all typical and atypical colonies were counted and tested, using a coagulase test for the confirmation of coagulase-positive staphylococci. Ten colonies of coagulase-positive staphylococci per cheese sample were used in further examinations.

**Bacterial strains**. A total of 180 coagulase-positive staphylococci were collected from 18 fresh cheese samples; 175 isolates were confirmed as *S. aureus*. Colonies were identified as S. aureus based on the coagulase test (Bactident^®^ Coagulase plasma EDTA, Merck, Darmstadt, Germany) and a latex agglutination test for the detection of fibrinogen affinity antigens, protein A, and capsular polysaccharides (PastorexTM Staph Plus test, Bio-Rad, Marnes-la-Coquette, France). All *S. aureus* isolates were tested for enterotoxin production and antibiotic susceptibility and their phenotypic characteristics were determined.

**Determination of phenotypic characteristics**. Methods used included Dnase (Dnase agar, Oxoid, Basingstoke, UK) and catalase tests, Gram staining, a haemolysis test (blood agar base No. 2 with defibrinated sheep blood, Oxoid, Basingstoke, UK), an “egg yolk” reaction on Baird-Parker agar (Oxoid, with Egg Yolk Tellurite Emulsion supplement, Oxoid, Basingstoke, UK), and biochemical identification using API STAPH (bioMerieux, Marcy-l’Etoile, France) according to manufacturer’s instructions.

**Antimicrobic susceptibility testing**. Antibiotic sensitivity was determined according to the EUCAST disk diffusion method [52] on Mueller–Hinton agar (Oxoid, Basingstoke, UK). The following antibiotic-impregnated disks were tested: azithromycin (15 µg), cefoxitin (30 µg), clindamycin (2 µg), erythromycin (15 µg), gentamicin (10 µg), moxifloxacin (5 µg), mupirocin (5 µg), oxacillin (5 µg), and trimethoprim-sulphamethoxazole (1.25/23.75 µg) (all Biorad, Marnes-la-Coquette, France). The interpretation of the results was conducted according to the European Committee on Antimicrobial Susceptibility Testing manual [52] and breakpoint tables for the interpretation of MICs and zone diameters, Version 13.1. [53].

**Hierarchical cluster analysis**. A hierarchical cluster analysis (HCA), a method of clustering to create a hierarchical tree or dendrogram, of the isolates was performed. This tree is a representation of the relationships between the objects, and it shows how the objects are grouped together into clusters at different levels of granularity. A Euclidean complete linkage hierarchical cluster analysis was conducted with the DataTab statistics calculator (DATAtab: DATAtab Team (2023). DATAtab: Online Statistics Calculator. DATAtab e.U. Graz, Austria. URL https://datatab.net accessed on 18 October 2023) which included, as metric variables, all data from the antimicrobic susceptibility testing and from the phenotypic identification. A dendrogram of all isolates was split into clusters.

**Detection of enterotoxin production in vitro**. All *S. aureus* isolates were examined for the production of SEA-SED enterotoxins. The overnight (37 °C/18–24 h) brain–heart infusion broth (Oxoid, Basingstoke, UK) cultures of well-isolated colonies from the Baird-Parker agar were examined using reversed passive latex agglutination RPLA (SET-RPLA, Thermo ScientificTM OxoidTM, Basingstoke, UK) methods, according to manufacturers’ instructions.

**DNA isolation**. Genomic DNA of each *S. aureus* isolate was extracted from one well-isolated colony on Baird-Parker agar in 100 µL of InstaGeneTM Matrix reagent (Bio-Rad, Marnes-la-Coquette, France) (1 colony in 100 µL of reagent). The suspension was vortexed for 15–20 sec and incubated in a dry incubator at 56 °C/60 min. After incubation, it was vortexed again for 15–20 sec and incubated for a second time at 95 °C/45 min. The final suspension was then vortexed for 15–20 min followed by centrifugation for 5 min at 13,200 rpm.

**Detection of SEA-SEE genes**. Since gene detection was performed using a modified real-time PCR method, classical PCR was used as a confirmatory method.

**Real-time PCR**. The real-time PCR method was a modified method described by Nakayama et al. [54] for the detection of sea-see genes. Primers (Invitrogen by Thermo Fisher Scientific, SAD) and probes (Applied Biosystems, Leicestershire, UK) are listed in Table 1. DNA amplification was conducted in a 20 μL reaction volume containing 2 μL DNA template and 18 μL reagent mix (10 μL of Luminaris Probe qPCR Master Mix (Thermo Scientific Baltics, Vilnius, Lithuania), 0.6 μL of each primer, 0.4 μL of MGB probe, and 6.4 μL of nuclease-free water). The real-time PCR reactions were carried out separately for each gene in a PikoReal 24 thermocycler (Thermo Fisher Scientific, Vantaa, Finland) under the following conditions: UDG pre-treatment at 50 °C/2 min and initial denaturation at 95 °C/10 min, followed by PCR amplification for 40 cycles at 95 °C/15 sec and 60 °C/60 sec.

**Classical PCR.** DNA amplification was conducted on a volume of 25 μL comprising 12.5 μL of RED Taq (10 mM of Tris HCl pH 8.3, 50 mM KCl, 1.5 mM MgCl2, 0.001% of gelatine, and 0.2 mM of each deoxyribonucleoside triphosphate), 0.5 µL (1 µM) of each primer, 5 μL of extracted DNA, and 6.5 μL of H_2_O. The presence of the *sea*, *seb*, *sec*, *sed*, and *see* genes was investigated. The primers and the amplification conditions used are given in Table 1. The PCR reaction was carried out in a Thermocycler Gene Amp, PCR System, 9700 Gold (Applied Biosystem, Foster City, CA, USA). The amplifications were analysed using an electrophoretic run on 1.5% agarose gel containing ethidium bromide (0.5 μg/mL); 10 μL of each PCR sample was loaded with 2 μL of 6× loading buffer (Fermentas, VWR Italy, Milan, Italy) and 5 μL of marker-PCR as reference DNA (Fermentas, VWR Italy, Milan, Italy). The run was carried out at a voltage of 100 V for about 1 h in TBE 10× (Trizma base, boric acid, and EDTA 0.5M pH 8). At the end of the run, the bands were viewed with a UV transilluminator (Fotodine 3-3102 Celbio, Milan, Italy). The primers and amplification conditions are listed in Table 2.

## 3. Results

A total of 180 coagulase-positive isolates were isolated from 18 cheese samples, and 175 were confirmed as *S. aureus* through latex agglutination and API STAPH tests. All isolates possessed phenotypic characteristics typical for *S. aureus* (protein A, bound and free coagulase, catalase, Dnase, and morphology). Variations were observed in certain biochemical reactions, including the “egg-yolk” reaction and haemolysis pattern (Table 3). The “Egg yolk” reaction was characteristic of 94 (53.7%) out of 175 isolates; 32 (18.3%) isolates showed a weak reaction, while in 49 (28%) isolates, the reaction was completely absent. Beta-haemolysis was observed in 103 (58.8%) isolates, while 64 (36.6%) isolates produced a double-haemolysis zone. On the other hand, 8 (4.6%) isolates were non-haemolytic. All isolates were split into 11 clusters for the hierarchical cluster analysis (Figure 1).

Antibiotic resistance was detected in 2 (1.1%) out of 175 *S. aureus* isolates, to mupirocin only.

Real-time PCR detected that 34 (19.4%) isolates were positive for the enterotoxin gene *sec* and 80.6% were not enterotoxigenic. Other classical genes were not detected. The presence of the *sec* gene was confirmed using the classical PCR method as well.

The production of the enterotoxin SEC was detected in all 34 (19.4%) isolates that carried the *sec* gene. Other classical enterotoxins were not detected (Table 4). 

The hierarchical cluster analysis of the 34 SEC-positive isolates is shown in Figure 2.

## 4. Discussion

According to De Buyser et al. [3] and Delmas et al. [4], raw milk cheeses are often contaminated with enterotoxigenic *S. aureus strains*. A previous study on the microbiological quality of fresh cheese produced in the Dubrovnik area, conducted by Ljevaković-Musladin et al. [50], is in agreement with those findings. That study showed that 80% (24/30 samples) of the examined fresh cheese samples were (highly) contaminated with *S. aureus* strains. At that time, studies on the enterotoxigenic potential of *S. aureus* isolates had not been conducted, and their enterotoxin production capability and associated risk remained unknown. Besides a study on virulence factors of *S. aureus* strains isolated from mastitic cow milk [56], there are no data on *S. aureus* strains isolated from food. In light of that, this study aimed at the phenotype and genotype profiling of *S. aureus* strains isolated from fresh cheese. 

Our *S. aureus* isolates from fresh cheese possessed typical phenotypic characteristics of *S. aureus*. However, variability was observed in the “egg-yolk” reactions and haemolysis patterns. Most of the isolates showed either characteristic or weak “egg-yolk” reactions, while in 28% of the isolates this reaction was absent. Unlike our results, other studies [25,40,57] have reported that the majority (more than 50%) of *S. aureus* isolates from mastitic cow milk and goat and sheep cheese were “egg-yolk”-negative. 

Most of our isolates were β-haemolytic or produced a double zone of α+β-haemolysis. These results are in agreement with other findings [22,24,33]. Jørgensen et al. [58] even reported that all *S. aureus* from cheese production in Norway was β-haemolytic. Akineden et al. [25] and Stephan et al. [57] reported double-haemolysis in the majority of *S. aureus* isolates from goat’s milk cheese in Germany and bovine mastitis in Switzerland. According to Hájek and Marsálek [59], β-haemolysis is a characteristic of *S. aureus* isolates of animal origin. 

The extremely low antimicrobic resistance of the *S. aureus* isolates in our study is in disagreement with other published data. Studies from Normanno et al. [23], Rola et al. [24], André et al. [60], Grispoldi et al. [61], and Papadopoulos et al. [62] reported the very high antimicrobic resistance of *S. aureus* isolates from milk and cheese to the antibiotics used in human and veterinary medicine. In our study, a small number of isolates were resistant to mupirocin, an antibiotic used in human medicine. According to Zhang et al. [46], among the 12 antibiotics, the resistance rates of penicillin and ampicillin were the highest worldwide, and the antibiotic resistance of ampicillin, gentamicin and chloramphenicol has increased over time. The results of a meta-analysis by Gajewska et al. [5] suggest that there is a low prevalence of MRSA in raw milk recovered from healthy animals, but it has increased in the past ten years.

Our study showed that *S. aureus* isolates from fresh cheese belonged to two genotypes/phenotypes: non-enterotoxigenic and sec genotype/SEC phenotype. The majority (80.6%) of *S. aureus* isolates from fresh cheese carried no SE genes. Similar results were reported by Hunt et al. [63]. The authors reported that 83.2% of *S. aureus* isolates from raw milk and raw milk cheeses in Ireland were not enterotoxigenic. However, we detected the *sec* gene in 34 (19.4%) isolates. All *sec*-gene-positive isolates produced SEC in vitro. Other studies have also shown that the *sec* gene is the most frequently found in *S. aureus* isolates from milk and raw milk cheese [20,25,27,28,29,30,31,32,33,35,40,41,46,64].

On the other hand, many authors reported, with high frequency, *sed* gene presence (alone or in combination with other genes) in *S. aureus* isolates from raw milk and raw milk cheeses [20,21,22,23,24,37,38,43,44,65].

Although *sea*, *seb*, and *sed* genes were not detected in our study, according to various studies, those genes are mostly found in *S. aureus* isolates from food implicated in staphylococcal food poisoning [14,15,16,17,18]. Rall et al. [39] reported the dominant presence of the *sea* gene in *S. aureus* isolates from raw milk in Brazil, while the presence of *sea* and *seb* genes were found in raw milk cheese in Turkey [66]. Several authors reported on the prevalence of the *sea* gene together with the *sed* gene in *S. aureus* isolates from raw milk and dairy products in Italy [21,22,23,36]. Unlike other classical enterotoxin genes, the *see* gene was rarely found in *S. aureus* isolates [12,16]. Only a few studies reported the presence of the *see* gene in *S. aureus* strains isolated from goat and sheep cheese [26] and in *S. aureus* isolates from mastitic cow milk in Italy [61].

Data on the enterotoxigenic potential of *S. aureus* isolates from fresh cheese in Croatia are scarce. The only study that determined the presence of SE genes in *S. aureus* isolates was conducted by Jaki Taklec [56]. The author reported the prevalence of *seg* and *sei* genes in 89.4% and 91.5%, respectively, and the presence of the *sec* gene in 44.7% of isolates. Other genes were not detected. These findings on the *sec* gene agree with our results. A meta-analysis conducted by Zhang et al. [46] reported that the highest detection rates for classical enterotoxins in *S. aureus* isolates from milk and dairy products between 1992 and 2021 worldwide were found for *sec* and *sea* genes.

However, in recent years, many studies on the enterotoxigenic properties of *S. aureus* isolates have found a high prevalence of newer enterotoxins [15,20,42,43,44,45,67].

The results from our study also showed that fresh cheeses were frequently contaminated with a heterogenous population of *S. aureus* strains, which were both enterotoxigenic and non-enterotoxigenic. The hierarchical cluster analysis (Figure 1 and Figure 2) confirmed the heterogenicity of the studied population. These observations agree with the findings of Loncarevic et al. [35], who found significant diversity in *S. aureus* isolates within single samples of raw milk and raw milk products. The authors also recommended testing up to ten isolates per sample to increase the chance of identifying a potential source of staphylococcal intoxication. The same principle was applied in our study.

The dominant production of SEC enterotoxin in *S. aureus* isolates from milk, dairy products (fresh cheese especially), and other food has also been reported in many studies worldwide. Similar results were found in Norway [31], Japan [32], Germany [25], Austria [29], Turkey [33], Italy [36], Switzerland [57], Ireland [63], France [68], the USA [69], and Spain [70]. According to these authors, the SEC enterotoxin is the most common enterotoxin produced by *S. aureus* strains isolated from animal samples and the most frequently found in cow and caprine milk as well as raw cow and caprine milk cheeses. It has also been recognised as an important cause of SFP associated with the consumption of dairy products [69]. According to the phenotypic characteristics of *S. aureus* isolates from our study, we assume that the isolates were of animal origin.

In comparison with our results, other studies reported the prevalence of other enterotoxins in milk, cheese, and dairy products. Borelli et al. [71] reported SEA, SEB, and SEC as the most common enterotoxin types in Canastra cheese in Brazil. Gonano et al. [29] and Normanno et al. [23] found that some *S. aureus* isolates from dairy products were able to produce more than one enterotoxin type, mostly SEA/SED and SEA/SEB/SEC combinations. SEB is the most frequent enterotoxin type from *S. aureus* isolates from sheep cheese in Slovakia [72]. Although the first studies on the enterotoxigenicity of *S. aureus* isolates from cheese in France showed that SEC is the most frequent enterotoxin type [68], later studies discovered a higher prevalence of enterotoxin SED-producing strains [34,37]. Morandi et al. [65] reported a high frequency of enterotoxins SEA and SED in raw milk and dairy products in Italy. Poli et al. [45] reported enterotoxin SED as the most frequent classical enterotoxin in Monte Veronese cheese in Italy. According to Grispoldi et al. [61], *S. aureus* isolates from mastitic cow milk are mostly SEA, SEE, and SED producers. SEA and SED are the most common causative agents of staphylococcal food poisoning associated with dairy products [65].

The prevalence of enterotoxigenic strains in fresh cheese from our study was found to be 19.4%. A similar frequency was found in Austria [29], Ireland [63], and France [68]. However, many studies on raw milk and raw milk cheeses have reported a higher prevalence of enterotoxigenic strains [31,35,36]. On the other hand, two extremes were reported by Borelli et al. [71] and Ertas et al. [66], as 93.3% and 2.3% of SE-producing strains of *S. aureus* were found in Brazilian Canastra cheese and sheep cheese in Turkey, respectively. 

The diversity of results between studies is due to the differences between *S. aureus* reservoirs in different countries, *S. aureus* ecovars, analytical methods with different sensitivity, and differences in number and type of analysed samples.

## 5. Conclusions

Most *S. aureus* isolates from our study were not enterotoxigenic. All isolates possessed the typical phenotypic characteristics of *S. aureus*. Antibiotic resistance was surprisingly low, and to mupirocin only. Enterotoxigenic potential was observed in 19.4% of isolates. Based on their enterotoxigenic profile, all isolates could be divided into two phenotypes/genotypes, non-enterotoxigenic and toxigenic SEC phenotype (*sec* genotype), since only SEC production and the *sec* gene were detected. Cheese samples were contaminated with a mixture of non-enterotoxigenic and SEC-producing strains.

Although the enterotoxigenic potential of *S. aureus* isolates from fresh cheese was lower than expected, it can significantly increase if there is horizontal gene transfer between enterotoxigenic and non-enterotoxigenic strains in the cheese population. Since certain strains possessed the *sec* gene and were also able to produce SEC in vitro, enterotoxin production in fresh cheese could not be ruled out.

## Figures and Tables

**Figure 1 microorganisms-11-02993-f001:**
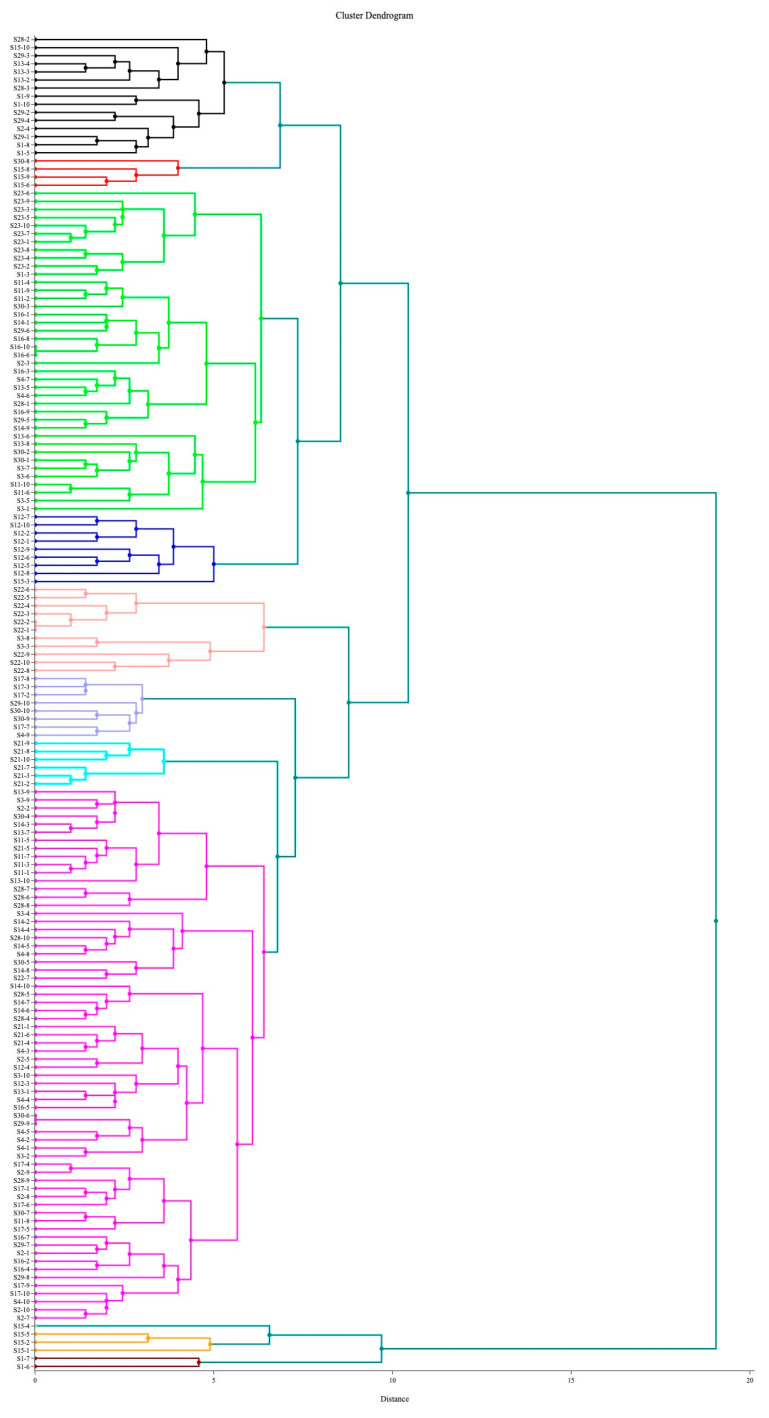
Cluster dendrogram for 175 *S. aureus* isolates.

**Figure 2 microorganisms-11-02993-f002:**
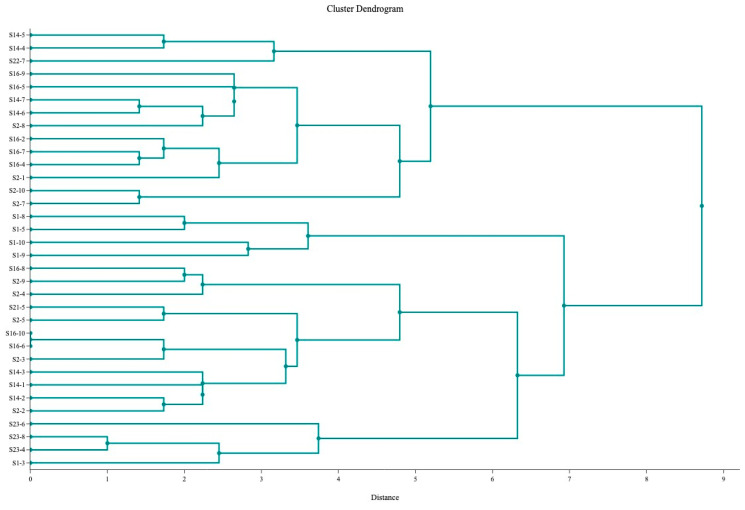
Cluster dendrogram for 34 SEC-positive *S. aureus* isolates.

**Table 1 microorganisms-11-02993-t001:** The primers and probes used in the real-time PCR.

Gene	Primer/Probe	Oligonucleotide Sequence (5′-3′) [54]	Position *	GenBank Accession no.
*sea*	eta-F	TTTGGAAACGGTTAAAACGAATAAG	489–513	M18970
	eta-R	TTTCCTGTAAATAACGTCTTGCTTGA	543–568	
	eta-T	FAM-CTGTTCAGGAGTTGGATC-MGB	524–541	
*seb*	etb-F	AGGTGACTGCTCAAGAATTAGATTACC	785–811	M11118
	etb-R	AAGGCGAGTTGTTAAATTCATAGAGTT	842–868	
	etb-T	FAM-AACTCGTCACTATTTGGTG-MGB	813–831	
*sec*	etc-F	GGCGATAAGTTTGACCAATCTAAATAT	811–837	X05815
	etc-R	AAGGTGGACTTCTATCTTCACACTTTT	864–900	
	etc-T	FAM-TGTACAACGACAATAAA-MGB	845–861	
*sed*	etd-F	CACAAGCAAGGCGCTATTTG	836–855	M28521
	etd-R	TCGGGAAAATCACCCTTAACA	966–986	
	etd-T	FAM-ATACAGCGCGGAAA-MGB	901–914	
*see*	ete-F	CTTTGGCGGTAAGGTGCAA	594–612	M21319
	ete-R	ACCGTGGACCCTTCAGAAGA	634–653	
	ete-T	FAM-AGGCTTGATTGTGTTTCA-MGB	615–632	

F—forward primer; R—reverse primer; FAM—6-carboxy-fluorescein; MGB—minor groove binder. * Positions correspond to the nucleotide numbers downstream from the ATG start codon of the respective enterotoxin genes.

**Table 2 microorganisms-11-02993-t002:** The primers used in the classical PCR.

Gene	Primer	Oligonucleotide Sequence (5′-3′) [54,55]	Amplification Conditions
*sea*	eta-F	AAAGTCCCGATCAATTTATGGCTA	94 °C 5′; 94 °C 3′; 58 °C 30″; 72 °C 5″; 30 cycles 72°C 10′
	eta-R	GTAATTAACCGAAGGTTCTGTAGA	
*seb*	etb-F	TCGCATCAAACTGACAAACG	94 °C 5′; 94 °C 2′ 55 °C 2′; 72 °C 1′; 30 cycles 72 °C 10′
	etb-R	GCAGGTACTCTATAAGTGCC	
*sec*	etc-F	GACATAAAAGCTAGGAATTT	94 °C 5′; 94 °C 2′ 55 °C 2′; 72 °C 1′; 30 cycles 72 °C 10′
	etc-R	AAATCGGATTAACATTATCC	
*sed*	etd-F	CTAGTTTGGTAATATCTCCT	94 °C 5′; 94 °C 2′ 55 °C 2′; 72 °C 1′; 30 cycles 72 °C 10′
	etd-R	TAATGCTATATCTTATAGGG	
*see*	ete-F	AGGTTTTTTCACAGGTCATCC	94 °C 5′; 94 °C 2′ 57 °C 2′; 72 °C 1′; 35 cycles 72 °C 7′
	ete-R	CTTTTTTTTCTTCGGTCAATC	

**Table 3 microorganisms-11-02993-t003:** Phenotypic characteristics of *S. aureus* isolates (n = 175).

	Coagulase	Latex Agglutination	Egg Yolk Reaction	Catalase	Dnase	Haemolysis
Positive	175	175	94 ^a^ + 32 ^b^	175	175	103 ^c^ + 64 ^d^
Negative	0	0	49	0	0	8

^a^—characteristic egg yolk reaction; ^b^—weak “egg yolk” reaction; ^c^—β-haemolysis; ^d^—double-haemolysis (α + β).

**Table 4 microorganisms-11-02993-t004:** Results of the detection of enterotoxin production in vitro using the RPLA method and the detection of classical *sea-see* genes using real-time PCR and classical PCR.

Sample	No. of Isolates	RPLA Phenotype	Real-Time PCR Genotype	Classical PCR Genotype
S1	7	SEC (5)	sec (5)	sec (4)
S2	9	SEC (9)	sec (9)	sec (9)
S3	10	Negative	Negative	-
S4	10	Negative	Negative	-
S8	10	Negative	Negative	-
S11	10	Negative	Negative	-
S12	10	Negative	Negative	-
S13	10	Negative	Negative	-
S14	10	SEC (7)	sec (7)	sec (5)
S15	9	Negative	Negative	-
S16	10	SEC (8)	sec (8)	sec (8)
S17	10	Negative	Negative	-
S21	10	SEC (1)	sec (1)	sec (1)
S22	10	SEC (1)	sec (1)	sec (1)
S23	10	SEC (3)	sec (3)	sec (3)
S28	10	Negative	Negative	-
S29	10	Negative	Negative	-
S30	10	Negative	Negative	-
Total	175	34	34	31

## Data Availability

Data from this study are available from the corresponding author upon request.

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
