# Peer review of "Enterotoxigenic and Antimicrobic Susceptibility Profile of *Staphylococcus aureus* Isolates from Fresh Cheese in Croatia"

_microorganisms, 2023, doi:10.3390/microorganisms11122993_

Round 1

Reviewer 1 Report (New Reviewer)

Comments and Suggestions for Authors

Subject: Review of Manuscript Submission to Microorganisms Journal - Major Revision Required 

Dear Authors,

I am writing to you in my capacity as a reviewer for the Microorganisms Journal regarding your manuscript entitled "Enterotoxigenic and antimicrobic susceptibility profile of Staphylococcus aureus isolates from fresh cheese in Croatia." First and foremost, I would like to commend you on the significance and relevance of your research. Your work delves into an area of paramount importance in microbiology and public health, particularly in understanding the characteristics and risks associated with Staphylococcus aureus in dairy products. The potential implications for food safety and antibiotic resistance outlined in your study are indeed noteworthy and contribute valuable insights to the field.

However, after a thorough review of your manuscript, I have concluded that it requires a major revision before it can be considered for publication. The research, as it stands, presents valuable data and findings, but there are several aspects that need to be addressed to enhance the clarity, depth, and impact of your work.

1. Introduction and Background: The introduction would benefit from a more detailed contextualization of your study within the existing body of research. Providing a clearer link between your objectives and the gaps or questions raised by previous studies would strengthen your rationale.

2. Materials and Methods: The methodology section, while detailed, requires additional clarity and justification for certain choices made in the study design, particularly in terms of sample selection and standardization of procedures.

3. Results: The presentation of results needs to be more comprehensive, particularly in the interpretation and significance of phenotypic variations and antimicrobial resistance findings.

4. Discussion: This section should offer a deeper analysis of how your findings support, extend, or contradict previous studies. The implications of your findings for public health and food safety need to be more thoroughly explored.

5. Conclusions: The conclusions drawn from your study should more explicitly state the public health implications and suggest directions for future research. A discussion on the likelihood and conditions for enterotoxin production and horizontal gene transfer is also warranted.

Your research addresses a critical area in food safety and microbiology, and with these suggested revisions, I believe it can make a significant contribution to the field. I recommend a major revision to address these concerns. Once these issues are addressed, I am confident that your manuscript will be a strong candidate for publication in the Microorganisms Journal.

Thank you for the opportunity to review this important work. I look forward to seeing the revised manuscript and the contributions it will undoubtedly make to our understanding of Staphylococcus aureus in food products.

Sincerely,

Please find my Review Report in the attached file.

Comments on the Quality of English Language

The English language in the document requires minor revisions for improved clarity and precision.

Author Response

Dear reviewers,

we hope this message finds you well. We sincerely appreciate the time and effort you dedicated to reviewing our manuscript entitled "Enterotoxigenic and Antimicrobic Susceptibility Profile of Staphylococcus aureus Isolates from Fresh Cheese in Croatia." Your detailed feedback has been invaluable, and we are grateful for the constructive insights you provided.

We are pleased to inform you that we have carefully considered each of your suggestions and have made significant revisions to address the concerns raised during your review. Here is a summary of the changes implemented in response to your feedback.

REVIEWER #1

  1. **Introduction and Background:**

- We have enhanced the contextualization of our study within existing literature to provide a clearer link between our objectives and the gaps identified in previous research.

  1. **Materials and Methods:**

- The methodology section has been revised for improved clarity and justification of choices made in study design, particularly in terms of sample selection and standardization of procedures.

  1. **Results:**

- The presentation of results has been made more comprehensive, with a focus on clearer interpretation and significance of phenotypic variations and antimicrobial resistance findings.

  1. **Discussion:**

- The discussion section now offers a deeper analysis of how our findings support, extend, or contradict previous studies. We have expanded on the implications of our findings for public health and food safety.

  1. **Conclusions:**

- The conclusions have been revised to more explicitly state the public health implications and suggest directions for future research. We have included a discussion on the likelihood and conditions for enterotoxin production and horizontal gene transfer.

In addition to addressing the overarching concerns, we have made specific adjustments based on the detailed points you raised in the Abstract and Introduction, Materials and Methods, Results, Discussion, and Conclusions sections. The revisions aim to improve the clarity, depth, and impact of our manuscript.

We believe that these revisions have significantly strengthened our manuscript, and we are confident that it now aligns more closely with the standards of the Microorganisms Journal. We hope that you find the updated manuscript satisfactory, and we look forward to receiving any further feedback you may have.

Once again, we appreciate your thoughtful review and the opportunity to enhance the quality of our work. Thank you for your time and consideration.

Beniamino Cenci Goga

Reviewer 2 Report (New Reviewer)

Comments and Suggestions for Authors

The manuscript « Enterotoxigenic and antimicrobic susceptibility profile of Staphylococcus aureus isolates from fresh cheese in Croatia” by Ljevkovic-Musladin et al, presents the phenotypical characterization of S. aureus strains isolated from 18 fresh cheeses.

180 strains were coagulase-positive staphylococci and 175 isolates were confirmed to be S. aureus. It is not clear how it was confirmed.  As presented in Table 3, it appears that 175 strains were coagulase-positive.

Why a simple PCR test was not performed to confirm S aureus?

Why some S. aureus strains were negative for Egg yolk reaction?

Some strains were mupirocin negative. However, this antibiotic was applied at 5 µg, while the new EUCAST disk are calibrated for 200 µg. Why 5 µg was used?

It is not clear why the presence of SEA-SEE genes was first tested by real-time PCR and, than, with PCR?  Both methods provided a qualitative result, while one would expect that real-time PCR was performed to quantify the presence of target genes. Why real-time PCR was done?

Author Response

Dear reviewers,

we hope this message finds you well. We sincerely appreciate the time and effort you dedicated to reviewing our manuscript entitled "Enterotoxigenic and Antimicrobic Susceptibility Profile of Staphylococcus aureus Isolates from Fresh Cheese in Croatia." Your detailed feedback has been invaluable, and we are grateful for the constructive insights you provided.

We are pleased to inform you that we have carefully considered each of your suggestions and have made significant revisions to address the concerns raised during your review. Here is a summary of the changes implemented in response to your feedback.

REVIEWER #2
The manuscript « Enterotoxigenic and antimicrobic susceptibility profile of Staphylococcus aureus isolates from fresh cheese in Croatia” by Ljevkovic-Musladin et al, presents the phenotypical characterization of S. aureus strains isolated from 18 fresh cheeses.

Question 1) 180 strains were coagulase-positive staphylococci and 175 isolates were confirmed to be S. aureus. It is not clear how it was confirmed. As presented in Table 3, it appears that 175 strains were coagulase-positive.

Answer 1) Table 3 lists only S. aureus (Table 3. Phenotypic characteristics of S. aureus isolates (n=175)

-------------------

Q2) Why a simple PCR test was not performed to confirm S aureus?

A2) We needed the biochemical tests for the clustering and opted for PCR only for SEs

-------------------

Q3) Why some S. aureus strains were negative for Egg yolk reaction?

A3) it is explained on lines 221-226 „S. aureus isolates from fresh cheese possessed typical phenotypic characteris7cs of S. aureus. However, variability was observed in „egg-yolk” reac7ons and hemolysis paderns. Most of the isolates showed either characteris7c or weak „egg-yolk” reac7ons, while in 28% of isolates this reac7on was absent. Unlike our results, other studies [24,29,50] reported that the majority (more than 50%) of S. aureus isolates from mas77c cow milk, goat, and sheep cheese were „egg-yolk“ nega7ve„

-------------------

Q4) Some strains were mupirocin negative. However, this antibiotic was applied at 5 μg, while the new EUCAST disk are calibrated for 200 μg. Why 5 μg was used?

A4) Resistant strains are commonly accepted as those with a MIC below 4mg/L (MIC ≤4mg/L) we defer to reviewer if the following references should be included in the discussion:

https://www.eucast.org/fileadmin/src/media/PDFs/EUCAST_files/Rationale_documents/Mupirocin_Rationale_Document_1.0.pdf

https://pubmed.ncbi.nlm.nih.gov/24144552/

-------------------

Q5) It is not clear why the presence of SEA-SEE genes was first tested by real-time PCR and, than, with PCR? Both methods provided a qualitative result, while one would expect that real-time PCR was performed to quantify the presence of target genes. Why real-time PCR was done?

A5) We explained this on lines 141-142. We defer to reviewer if this should be stated clearer in the text„Detection of SEA-SEE genes. Since gene detection was performed by modified Real-%me PCR method, classical PCR was used as confirmatory method“.

We believe that these revisions have significantly strengthened our manuscript, and we are confident that it now aligns more closely with the standards of the Microorganisms Journal. We hope that you find the updated manuscript sa7sfactory, and we look forward to receiving any further feedback you may have.

Once again, we appreciate your thoughful review and the opportunity to enhance the quality of our work. Thank you for your 7me and consideration.

Beniamino Cenci Goga

Round 2

Reviewer 1 Report (New Reviewer)

Comments and Suggestions for Authors

Dear Authors, 

I hope this letter finds you well. I am writing to you in my capacity as a reviewer regarding the manuscript titled “Enterotoxigenic and antimicrobic susceptibility profile of Staphylococcus aureus isolates from fresh cheese in Croatia” with ID [microorganisms-2731097], which you have submitted to Microorganisms Journal.

Firstly, I would like to express my appreciation for the effort and time you have invested in revising your manuscript based on the initial round of reviews. However, after a thorough examination of your revised submission, I have noticed that a detailed, point-by-point response to the reviewers' comments has not been provided. This step is crucial as it helps the reviewers and editorial team understand how you have addressed each specific concern raised in the initial review.

Furthermore, it is standard practice to highlight changes made in the revised manuscript. This not only facilitates a smoother and more efficient review process but also ensures transparency in how the feedback has been incorporated. A clean version of the manuscript should also be provided alongside the highlighted version for clarity and ease of reading.

Given these considerations, I regret to inform you that we will need to go through a second round of revision. I kindly ask that you:

  1. Provide a detailed, point-by-point response to each of the reviewers' comments.
  2. Highlight the changes made in the revised manuscript to clearly indicate where modifications have been made.
  3. Include a clean version of the manuscript for ease of evaluation. 

We believe these steps are essential for a thorough and fair assessment of your work. Please ensure that these requirements are met in your resubmission.

We appreciate your understanding and cooperation in this matter. Our aim is to maintain the highest standards of academic rigor and integrity, and your compliance with these revision guidelines will greatly contribute to this goal.

Please feel free to contact me if you have any questions or require further clarification regarding this process. We look forward to receiving your revised manuscript.

Thank you for your attention to this matter.

Sincerely,

Comments on the Quality of English Language

English language needs minor editing.

Author Response

Please find enclosed our answers.

Reviewer 2 Report (New Reviewer)

Comments and Suggestions for Authors

Authors answerd all my concerns. 

Author Response

We would like to thank reviewer #2 for the swift and professional evaluation

This manuscript is a resubmission of an earlier submission. The following is a list of the peer review reports and author responses from that submission.

Round 1

Reviewer 1 Report

Comments and Suggestions for Authors

The manuscript reports the identification of Staphylococcus aureus isolates from fresh cheese samples in Croatia.

The study aim is interesting as similar reports from Croatia are very limited. 

However, the experimental design followed raises some concerns on the possible overestimation of some of the results obtained.

- Ten isolates were collected per cheese sample, yet no typing of the isolates was made. Without this information, or an in-depth phenotypic and genotypic characterization of the isolates it is not possible to guarantee that the isolates collected from the same sample represent the same strain or different strains. This may lead to a very significant overestimation of the frequency of positive isolates/strains. For the same reason, there is insufficient support for the statement in Lines 247 - 249, in which authors state that the fresh cheeses were contaminated with a heterogeneous population of S. aureus strains.

- Antibiotic susceptibility was tested according to EUCAST guidelines. However, there is no reference to the clinical breakpoints used to categorize the isolates as susceptible, intermediate or resistant. There are also some concerns regarding the interpretation of antibiotic susceptibility data, particularly for mupirocin. EUCAST presents an ECOFF for mupirocin based on disks with a content of 200 ug of mupirocin. The use of disks with 5 ug of mupirocin as was used in this study, would impair the categorization of the isolates as S/I/R.

- Some other aspects could also improve the sudy such as the inclusion of additional enterotoxins in the screening.

Comments on the Quality of English Language

The manuscript is in general well written and organized.  

Reviewer 2 Report

Comments and Suggestions for Authors

The paper is written within the Journal required style. The whole study was seriously conducted and the methodology applied is correct (with the exceptions described below). The manuscript is well-designed, results and discussion are plausible and coherent according to the scientific standards; they are also explicatory and sufficiently presented with several tables. The objective of the study is well-defined and the Authors explain how they reached the conclusions. The paper is also written quite correctly in terms of its language.

The only comment I have concerns the number of bacteria (isolates or staphylococci) investigated. Authors should highlight and explain what they mean by the term "isolates". In my opinion, the authors mean bacterial colonies, and in this situation, the term isolates is misleading. Furthermore, the number of colonies (isolates) obtained from a given cheese should not be mentioned. The authors should note that this is a number obtained from a specific mass of material (quantitative analyzes were performed). In this situation, the result obtained is usually given as CFU. How should the terms "typical and atypical colonies" be understood? This information should also be clarified. How were 10 colonies selected from a given cheese? Table 4 shows that it was not always 10 colonies. And, in addition, the centrifugation speed is better reported as RCF.